# Automated Analysis of Platelet Aggregation on Cultured Endothelium in a Microfluidic Chip Perfused with Human Whole Blood

**DOI:** 10.3390/mi10110781

**Published:** 2019-11-14

**Authors:** Hugo J. Albers, Robert Passier, Albert van den Berg, Andries D. van der Meer

**Affiliations:** 1BIOS Lab-on-a-Chip Group, University of Twente, 7522 NH Enschede, The Netherlands; 2Applied Stem Cell Technologies Group, University of Twente, 7522 NB Enschede, The Netherlands

**Keywords:** triangle threshold, platelet aggregate, platelet coverage, platelet size distribution, automatic threshold, image analysis, thrombosis, thrombosis-on-a-chip, microfluidics, organ-on-a-chip

## Abstract

Organ-on-a-chip models with incorporated vasculature are becoming more popular to study platelet biology. A large variety of image analysis techniques are currently used to determine platelet coverage, ranging from manually setting thresholds to scoring platelet aggregates. In this communication, an automated methodology is introduced, which corrects misalignment of a microfluidic channel, automatically defines regions of interest and utilizes a triangle threshold to determine platelet coverages and platelet aggregate size distributions. A comparison between the automated methodology and manual identification of platelet aggregates shows a high accuracy of the triangle methodology. Furthermore, the image analysis methodology can determine platelet coverages and platelet size distributions in microfluidic channels lined with either untreated or activated endothelium used for whole blood perfusion, proving the robustness of the method.

## 1. Introduction

Cardiovascular diseases are the leading cause of death worldwide [1]. A lot of research has already been conducted to better understand, prevent and treat cardiovascular diseases. Animal models are frequently used as disease models [2], but often do not fully recapitulate human physiology [3,4]. Organ-on-a-chip models with integrated vascular compartments are promising *in vitro* alternatives that allow for control over used geometries and applied flow profiles. These models can be lined with human endothelial cells, only require small amounts of human whole blood for perfusion and, most importantly, allow for complex, dynamic aspects of cardiovascular diseases to be studied in an all-human setting [5,6,7,8,9,10,11,12,13,14].

Image analysis is an integral part of studying the behavior of cells in tissue culture *in vitro*, and multiple software packages exist to determine the confluency of a cell monolayer (with ImageJ) [15], track cell migration (with CellProfiler) [16], or monitor smooth muscle cells or cardiomyocyte contraction (with MUSCLEMOTION) [17]. Similarly, when incorporating blood-derived platelets and leukocytes into organs-on-chips, robust methods are needed for quantifying key descriptive indicators or characteristics, e.g., rolling, adhesion, clotting time, platelet coverage and aggregate size distribution. Furthermore, a reliable image analysis technique suitable for organs-on-chips is required before these models can be used in personalized medicine applications or implemented in the drug development pipeline [18].

To quantify platelet adhesion in microfluidic channels or flow chambers, various strategies have been applied. When using fluorescently labeled platelets, the mean intensity of representative areas of a channel or chamber can be used as a measure of platelet coverage and aggregate growth [5,6,19]. However, this strategy only gives an averaged signal and does not allow for discrimination of single platelets or individual aggregates. In order to obtain data on individual platelets or aggregates, alternative strategies have been developed to manually select individual platelets in flow chambers [20] or to assign a morphological score to regions of interest (ROIs), differentiating between a few single adhered platelets to large platelet aggregates [21]. The downside of such methods is that they rely on manual selection and scoring, thereby limiting the throughput and leaving room for human error or bias. In order to automatically obtain information on platelet coverage [8,11,21,22,23,24] and growth dynamics [22,23,25], a threshold will have to be applied to fluorescence microscopy data to differentiate platelet aggregates from background signals [8,11,21,24]. The level of the threshold can be set manually by a researcher to obtain the best signal-to-background ratio [21,25], but this again introduces issues related to the throughput and bias. The threshold can also be automatically determined using various automatic threshold techniques, e.g., mean, Otsu and triangle. These techniques determine a threshold by measuring the mean intensity (mean), minimizing intra-class variance in a bimodal histogram (with the Otsu method) [26] or by locating the base of a histogram peak (with the triangle method) [27]. Given the nature of the fluorescence microscopy data, with sparse areas of positive signal on a mostly negative background, resulting in a skewed unimodal histogram, the triangle method [27] stands out. Variations of this method have been applied successfully for platelet coverage analysis in collagen-coated flow chambers [22,23]. However, the method has not been applied to study platelet aggregation patterns in microfluidic channels lined by endothelial cells, which are an integral part of organs-on-chips.

When applying automated platelet analysis in microfluidic channels, edge effects have to be taken into account [28]. Low shear rates close to the channel edges might exacerbate fibrin deposition [8] and including these regions can therefore result in non-representative platelet coverage values. Moreover, endothelial cells in microfluidic channels can be either in a non-activated state which almost fully prevents platelet adhesion and aggregation, or in an activated state with dense platelet aggregates stabilized by fibrin [6,8,11]. It is unknown if the previously reported methods for automated analysis can be applied to reliably quantify platelet aggregation patterns in microfluidic channels lined by endothelial cells in activated and non-activated states.

In this communication, we introduce an image analysis methodology to quantify platelet aggregate coverage and size distribution in microfluidic devices lined with activated and non-activated endothelial cells and perfused with human whole blood at arterial shear rates. We demonstrate that image analysis of unprocessed data can be automated by correcting channel misalignment, creating ROIs, extracting areas covered by platelet aggregates and determining their size distribution. Here, we show, for the first time, that this triangle methodology is suitable for the characterization of both non-activated and activated endothelium cultured in microfluidic channels. Furthermore, the triangle methodology is compared to manual identification of single platelets and platelet aggregates and shows high degree of sensitivity, specificity and accuracy.

## 2. Materials and Methods 

### 2.1. Microfluidic Device Fabrication

Polydimethylsiloxane (PDMS, Sylgard 184, Dow Corning, Midland, MI, USA) channels were made using standard soft lithography techniques. First a mould was fabricated by patterning a negative photoresist (SU-8, Microchem, Round Rock, TX, USA) on a silicon wafer using photolithography. Then, a PDMS base and a curing agent were mixed at a 10:1 ratio (*w*/*w*) degassed, poured and baked at 60 °C for 4 h. The patterned PDMS slab was peeled off and trimmed, and its inlets and outlets, both 1 mm in diameter, were punched (Integra Miltex Biopsy Punch). The PDMS was bonded to a glass slide (Thermo Fisher Scientific, Waltham, MA, USA) using a plasma treatment (50 W, 50 kHz, 0.5 Torr for 40 s, CUTE MPR, Femto Science), which resulted in a sterile microfluidic channel of 51 µm × 300 µm × 14 mm (height × width × length, Figure 1B). 

### 2.2. Endothelial Cell Seeding and Introduction of Inflammatory Cytokines

Directly after the plasma treatment, a 0.1 mg/mL collagen-I (Corning rat tail collagen-I, high concentration) solution was incubated at 37 °C for 30 min. Human umbilical vein endothelial cells (HUVECs, Lonza C2519A) were trypsinized (Gibco), centrifuged at 390 ×g for 5 min and seeded twice at a seeding density of 15 × 10^6^ cells/mL in endothelial cell growth medium (ECGM, Cell Applications, San Diego, CA, USA) to cover both the bottom and top of the microfluidic channel [8]. Cells were allowed to reach confluency before introducing the ECGM with (activated condition) or without (non-activated condition) 10–50 ng/mL tumor necrosis factor-α (TNF-α, Sigma Aldrich, St. Louis, MO, USA) and being incubated overnight. Images visualizing endothelial cell morphology were made by fixating the HUVEC monolayer with a 4% (*v*/*v*) formaldehyde (Thermo Fisher Scientific) solution followed by a permeabilization with 0.3% (*v*/*v*) Triton-X-100 (Sigma Aldrich, St. Louis, MO, USA). The nuclei and actin filaments were stained by using NucBlue and ActinGreen (Thermo Fisher Scientific) and following the manufacturer’s protocol, and imaged using DAPI and GFP filter cubes.

### 2.3. Whole Blood Perfusion

Using a syringe pump (Harvard PHD 2000 Syringe Pump), custom connectors and tubing (Tygon) ECGM perfusion was established at an arterial shear rate of 1000 s^−1^. Human whole blood was provided by the Experimental Centre for Technical Medicine from the Technical Medical Centre, University of Twente, and used within 4 h after being drawn (3.2% sodium citrate, Vacuette, Greiner Bio-One). This research did not fall in the scope of the Dutch Medical Research Involving Human Subjects Act. In agreement with the Declaration of Helsinki, informed consent was obtained from all volunteers. Furthermore, the blood collection procedure was approved by the local medical research ethics committee (METC Twente). The platelets in the whole blood were stained using CD41-PE (1% (*v*/*v*), Thermo Fisher Scientific) or DiOC_6_ (1 µg/mL, Thermo Fisher Scientific). After incubating the label, the citrated human whole blood was reconstituted by adding a re-calcification buffer (Gibco HEPES, 63.2 mM CaCl_2_ from Sigma Aldrich, St. Louis, MO, USA and 31.6 mM MgCl_2_ from Ambion). The re-calcified citrated whole blood with stained platelets was perfused at 1000 s^−1^ for 25 min. The volumetric flow rate was determined using Equation (1): (1)$$\gamma =6Q/wh^{2},$$
where *γ* is the shear rate (s^−1^), *Q* is the volumetric flow (m^3^/s), *w* is the channel width (m), and *h* is the channel height (m).

After the whole blood perfusion, the channel was washed with ECGM followed by a 4% (*v*/*v*) formaldehyde (Thermo Fisher Scientific, Waltham, MA, USA) wash to fixate the cells. Fluorescence microscopy images and phase contrast images were acquired using the EVOS FL imaging system (Thermo Fisher Scientific, Waltham, MA, USA). A 10× magnification was used for all images. To measure platelet coverage in the microfluidic devices, an RFP filter cube was used when platelets were stained using CD41-PE and a GFP filter cube was used for samples stained with DiOC_6_.

### 2.4. Determination of an ROI and Triangle Thresholding

Unprocessed sets of fluorescence microscopy images were imported in MATLAB (MathWorks), corrected for channel misalignment and two ROIs were created per figure. ROI #1 is the area spanned by the microfluidic channel and ROI #2 is the area that is located at least 50 µm from channel walls and figure borders. Using the triangle method, a threshold value was found in ROI #1 and applied in ROI #2, resulting in a black and white image where the white areas represent platelets or platelet aggregates. Platelet coverage was measured by calculating the ratio of white pixels versus black pixels. Platelet aggregate size distributions were determined by characterizing the white areas in the binary figures. Two scripts for determining platelet coverages and platelet aggregate size distributions have been released on the MATLAB central file exchange and can be found in the Appendix A.

### 2.5. Manual Identification of Platelet Aggregates and Statistics

ROI #2 was manually processed by identifying individual platelets and platelet aggregates in MATLAB. The black and white images resulting from the manual identification of platelets and platelet aggregates were compared to the black and white images from the triangle methodology. A direct comparison resulted in true positives, false positives, true negatives and false negatives used to calculate sensitivity, specificity and accuracy. After calculating platelet coverages, using the same method as mentioned in Section 2.4, the platelet coverages found with the triangle methodology were plotted and compared to the platelet coverages found with the manual identification method. Furthermore, the R-squared value was determined by calculating squared residuals compared to *y* = *x* using MATLAB.

### 2.6. Computational Fluid Dynamics

COMSOL Multiphysics 5.4 was used to conduct computational fluid dynamics modeling of the wall shear rate in the microfluidic device. A 2 mm long section of the microfluidic channel was modeled using the laminar flow module for incompressible flow. A no-slip boundary condition was imposed on all walls and a volumetric flow rate of 7.8 µL/min was applied on the inlet, while the atmospheric pressure was maintained on the outlet. Shear rate profiles were mapped on the three-dimensional (3D) model and the cutline data were exported to MATLAB for visualization.

## 3. Results

### 3.1. Microfluidic Device and Introduction of Data

Microfluidic chips were used for human whole blood perfusion experiments in endothelialized channels. A fluorescence microscopy image of a confluent monolayer of HUVECs is shown in Figure 1A (nuclei in blue and F-actin in green). The HUVEC monolayer was left untreated or exposed to TNF-α overnight to instigate inflammation. After 25 min of whole blood perfusion, the channels were rinsed and fixated. Figure 1 shows the used microfluidic chip and typical phase contrast and fluorescence microscopy images.

Both phase contrast data and fluorescence data can be used to measure platelet aggregation. For the phase contrast data, either the platelets or platelet aggregates have to be selected manually [20] or detected automatically using edge detection. Manually selecting adhered platelets (Figure 1C,E) and platelet aggregates is slow and prone to human error. The edge detection method should pick up not only single platelets but also platelet aggregates, which might have similar sizes and shapes compared to other particles like red blood cells, white blood cells, apoptotic cells and apoptotic bodies [29]. The use of fluorescence data (Figure 1D,F) is more robust and circumvents any accidental non-specific detection. A threshold can also be automatically determined by various automatic threshold techniques. In ImageJ (NIH Image) [15], 16 automated thresholds were qualitatively compared using a set of representative fluorescence microscopy images. The Otsu and triangle methods were the best at distinguishing platelets from the background. The Otsu method finds the threshold that minimizes the intra-class variance and works best with bimodal histograms [26]. However, minimal platelet adhesion is expected on non-activated endothelium, resulting in a unimodal histogram which makes the Otsu technique less suitable. The triangle method is a geometrical threshold method aimed at setting a threshold at the base of a histogram peak and works best for a skewed unimodal histogram [27]. The fluorescence microscopy images of adhered platelets have a unimodal histogram, and therefore, the triangle method is suitable for determining the threshold.

### 3.2. Automated Threshold Using the Triangle Method

Fluorescence microscopy images were imported into MATLAB (version R2016b). To correct for misalignment of the microfluidic channel on the microscopy stage, the top edge of the channel was found by manually indicating the intersection of the top wall of the channel and the left and right borders of the figure (the top dashed line in Figure 2A). Using the coordinates of these intersections and the arctangent, the angle between the channel walls and the true horizontal line was calculated and corrected using the “imrotate” function in MATLAB. Alternatively, the channel edge was found automatically by vertically scanning the image to find the first local maximum intensity followed by an angle sweep to determine the misalignment angle. Using the angle-corrected figure and the coordinates of the top channel wall extremities, two ROIs were cropped: ROI #1 spans the entire area inside the channel walls, and ROI #2 spans the area inside the channel at least 50 µm from the channel walls and image borders, as indicated in Figure 2B. Because rectangular channels were used, the flow rate close to the channel was affected by edge effects, resulting in lower wall shear rates. The area spanned by ROI#2 shows a flat wall shear rate profile and displays shear rates of ≥95% of the maximum value (Figure A1, Appendix B). Therefore, the area located up to 50 µm from the channel wall was omitted in platelet coverage calculations.

The two ROIs were used to determine the platelet coverage and aggregate size distribution using the triangle threshold method. For the triangle threshold method to find a threshold that distinguishes adhered platelets from background, there need to be platelet aggregates. In the absence of foreground signals, the calculated threshold will shift and include a large proportion of the background signal, resulting in an exaggerated platelet coverage value. When analyzing microfluidic channels with healthy endothelium, little to no platelet adhesion was expected in ROI #2, while platelet aggregation can be expected in a 50 µm outer rim of ROI #1, due to edge effects. By using ROI #1 to determine the threshold and thus including aggregates formed in the proximity of the channel walls, as well as applying this threshold in ROI #2, this problem was circumvented.

The histogram of ROI #1 was used to determine the threshold by finding the maximum count and the artificial zero count (Figure 3A). This artificial zero was set by finding the first bin, in which the count was lower than 0.01% of the maximum count of the histogram. A linear line, the hypotenuse, was drawn from the artificial zero to the maximum count (Figure 3B). For each bin, the shortest distance between the hypotenuse and the histogram was calculated, which by definition was the length of a line orthogonal to the hypotenuse. The bin with the maximum distance between the histogram and the hypotenuse was set as the threshold (Figure 3C). Finally, the found threshold was imposed on ROI #2 (Figure 3D), resulting in a binary image where platelets and platelet aggregates are shown in white (Figure 2C).

Platelet coverage was determined by dividing the number of non-zero (white areas) values by the number of zero values × 100. The binary images were used to investigate the aggregate size distribution by using the “regionprops” function in MATLAB.

### 3.3. Comparison of Binary Output Using the Triangle Method with the Manual Method

The proof of concept in Figure 2 shows that the triangle threshold is suitable for distinguishing platelets from the background. To determine the performance of this method in finding platelet aggregates, 10 fluorescence microscopy images were processed with the triangle threshold script and were compared to manually processed images. Higher-intensity versions of the fluorescence microscopy images were processed by manually drawing freeform lines around platelet aggregates, creating a mask representing the location and size of platelet aggregates.

The two black and white images (Figure 4B,C), produced with the triangle threshold and the manually drawn mask, show similarly shaped aggregates and also the measured platelet aggregate coverage is quite similar: 3.36% for the triangle threshold and 2.96% for the manual method. Both platelet aggregate size distributions were compared and displayed similar distributions (Figure A2, Appendix B). To determine how the triangle methodology compared to the manual identification, masks from both techniques were superimposed and a sensitivity of 84.84%, a specificity of 98.69% and an accuracy of 91.27% were determined. The triangle methodology recognized true positive values (sensitivity) and was even better at identifying true negatives (specificity). Furthermore, the triangle methodology was capable of detecting both true positives and true negatives, resulting in a high accuracy. The accuracy of each individual image is given in Figure 4D, showing high accuracies for untreated and activated endothelium. A comparison of the platelet coverages determined using the triangle methodology and manual identification is shown in Figure 4E, where the coordinates of manually determined coverages are plotted versus triangle-determined coverages (*R*^2^ = 0.9915). Figure 4D,E show data generated with microfluidic channels lined with untreated and activated endothelium. The triangle method is capable of handling widely varying data ranging from almost no platelet coverage for untreated channels to occluded channels lined with activated endothelium. This versatility shows the potential of using a triangle threshold methodology for the analysis of platelet aggregates.

To further illustrate the robustness of our method, we also analyzed fluorescence data that were obtained from experiments, in which the platelets were labeled with DiOC_6_ instead of CD41-PE. DiOC_6_ targeted mitochondria and, when added to a blood sample, stained endothelial cells over the course of the blood perfusion experiment. This resulted in an increase in background signal over time, but we demonstrated that patterns of DiOC_6_ labeled platelets can also be successfully analyzed using the triangle method (Figure A3, Appendix B).

## 4. Discussion

We have demonstrated that the triangle threshold method is a suitable way to analyze patterns of platelets in microfluidic channels lined with endothelial cells. The method correlates well with manual processing, detects platelet aggregates in a wide size range and is robust enough to deal with data with extremely low platelet coverage and high platelet coverage on healthy and activated endothelium, respectively.

The expected platelet adhesion patterns on endothelial cells range from single adhered platelets to 3D blood clots. The script only measures areas covered by platelets and thus the volumetric 3D platelet aggregates are collapsed onto an area. Using confocal microscopy might be of interest to determine whether this simplification is valid and might elucidate a link between platelet aggregate size and volume, adding to the descriptive power of the current methodology.

Others have calculated the platelet coverage only including thresholded areas larger than a single platelet [23]. Only microscopy images that are in the focus of the endothelial monolayer should be used, but because of the heterogeneous nature of the aggregates (small aggregates and bigger 3D aggregates), a large 3D aggregate might be partially out of focus, resulting in a slightly granular thresholded area. The out of focus area also represents an area covered by platelets and should be used in coverage measurements.

Studying platelet aggregation in flow chambers has been performed for decades and has led to important insights into platelet biology. Microfluidic channels coated with extracellular matrix proteins have helped understand bleeding and vascular injury [22,23]. Recently, organs-on-chips and their integrated blood vessels-on-chips are becoming increasingly popular to study platelet biology. These vessel-on-chip devices incorporate endothelial cells which opens up opportunity to study the interaction between endothelium and whole blood [5,6,7,8,9,10,11,12,13,14]. However, increased complexity goes hand in hand with challenges in analyzing data. The introduced methodology could be used by others to automatically determine platelet coverage and platelet aggregate size distributions in their organ-on-a-chip devices, even for data analysis of experiments with widely varying platelet coverages.

## 5. Conclusions

In this communication, an automated analysis of platelet coverage and platelet size distribution was introduced for applications in organs-on-a-chips and vasculature-on-a-chip devices. The methodology corrects for channel misalignment, automatically defines ROIs and sets a threshold using the triangle method. The described method was compared to a manual identification method, where a user manually indicates adhered platelets and aggregates. Furthermore, a high sensitivity, a high specificity and a high accuracy were measured. The image analysis method presented here is capable of determining platelet coverages and platelet size distributions in microfluidic devices perfused with human whole blood lined with either activated or untreated endothelial cells, proving the robustness of the methodology.

## Figures and Tables

**Figure 1 micromachines-10-00781-f001:**
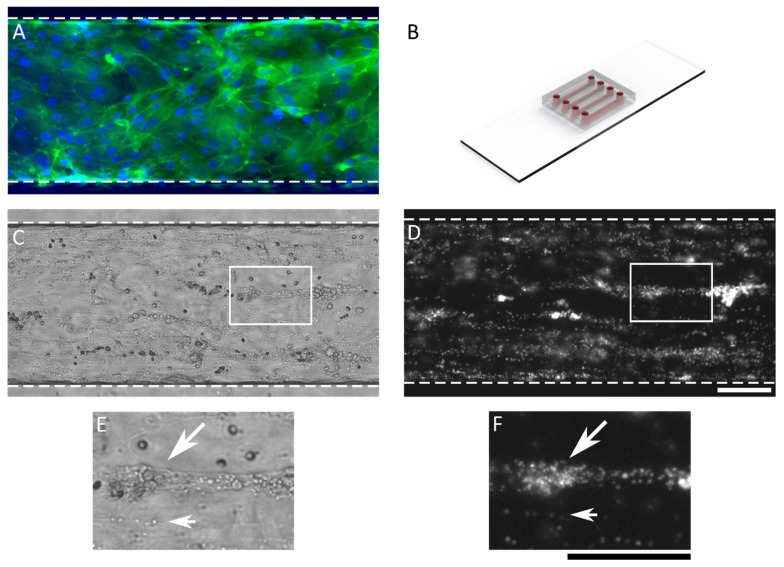
Microfluidic chip and microscopy images. (**A**) Fluorescence microscopy image of a confluent monolayer of human umbilical vein endothelial cells (HUVECs) in a microfluidic chip (nuclei are in blue; F-actin is in green). (**B**) Render of the microfluidic chip used. Channels have dimensions of 51 µm × 300 µm × 14 mm (height × width × length). (**C**,**E**) Phase contrast images of the microfluidic channel lined with endothelial cells after fixation and 25 min of whole blood perfusion. (**D**,**F**) Fluorescence microscopy image of platelets stained with CD41-PE and fixated after 25 min of whole blood perfusion. Insets in (**E**,**F**) show small platelet aggregates (large arrows) and single adhered platelets (small arrows). Scale bars, 100 µm.

**Figure 2 micromachines-10-00781-f002:**
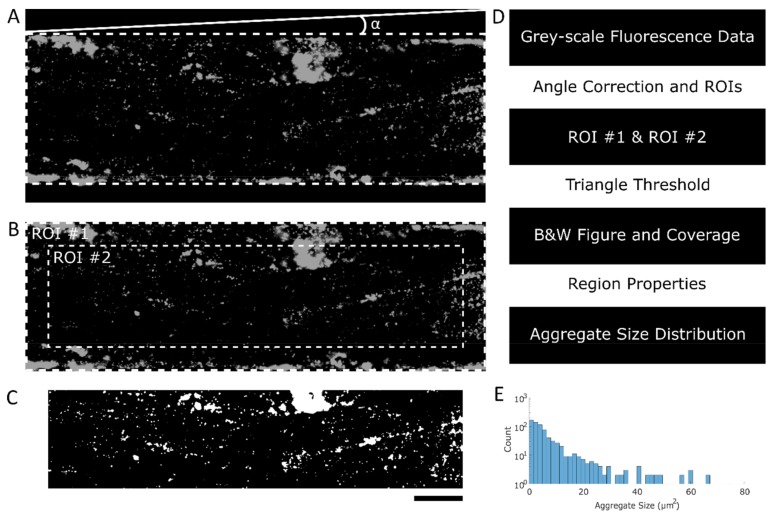
Conversion of a grey-scale fluorescence microscopy image to a black and white image. (**A**) Raw data showing platelet aggregates in greyscale. The angle α indicates possible channel misalignment. (**B**) Angle-corrected and -cropped image. Dashed lines indicate the regions of interest. ROI #1 spans the entire microfluidic channel, and ROI #2 is the area of the microfluidic channel minus an outer rim of 50 µm. (**C**) Resulting black and white image after the triangle threshold was applied. (**D**) Schematic representation of script operations and the resulting outputs. (**E**) Platelet aggregate size distribution. Scale bar, 100 µm.

**Figure 3 micromachines-10-00781-f003:**
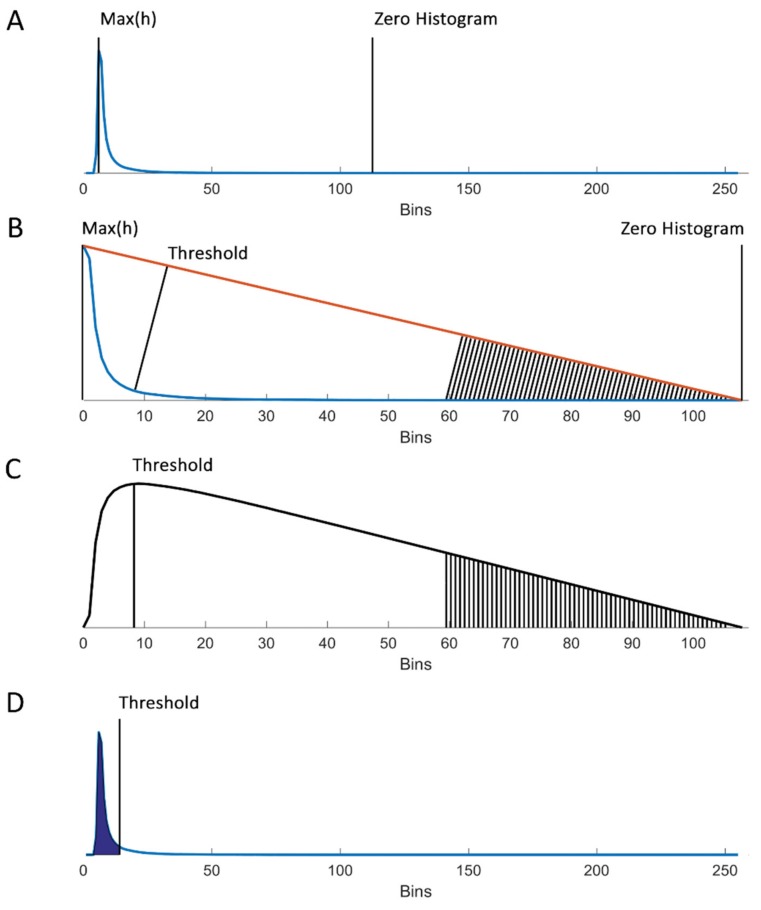
Determination of the threshold in the fluorescence intensity histograms of platelet aggregation data with the triangle method. (**A**) Histogram of raw data with “bins” of fluorescence intensity on the *x*-axis and frequency on the *y*-axis. The maximum and the artificially set zero points are indicated. (**B**) The processed histogram in which the first bin coincides with the maximum count bin and the last bin coincides with the zero histogram bin. The orange line is the hypotenuse, connecting the maximum count bin in the histogram (“Max(h)”) and the zero histogram bin. The shortest distance between the histogram and the hypotenuse was measured for each bin, which by definition was the length of the line orthogonal to the hypotenuse. (**C**) The resulting distances between the hypotenuse and histogram for each bin of the histogram. The bin with the maximum distance was chosen as the threshold. (**D**) Histogram with the triangle threshold applied.

**Figure 4 micromachines-10-00781-f004:**
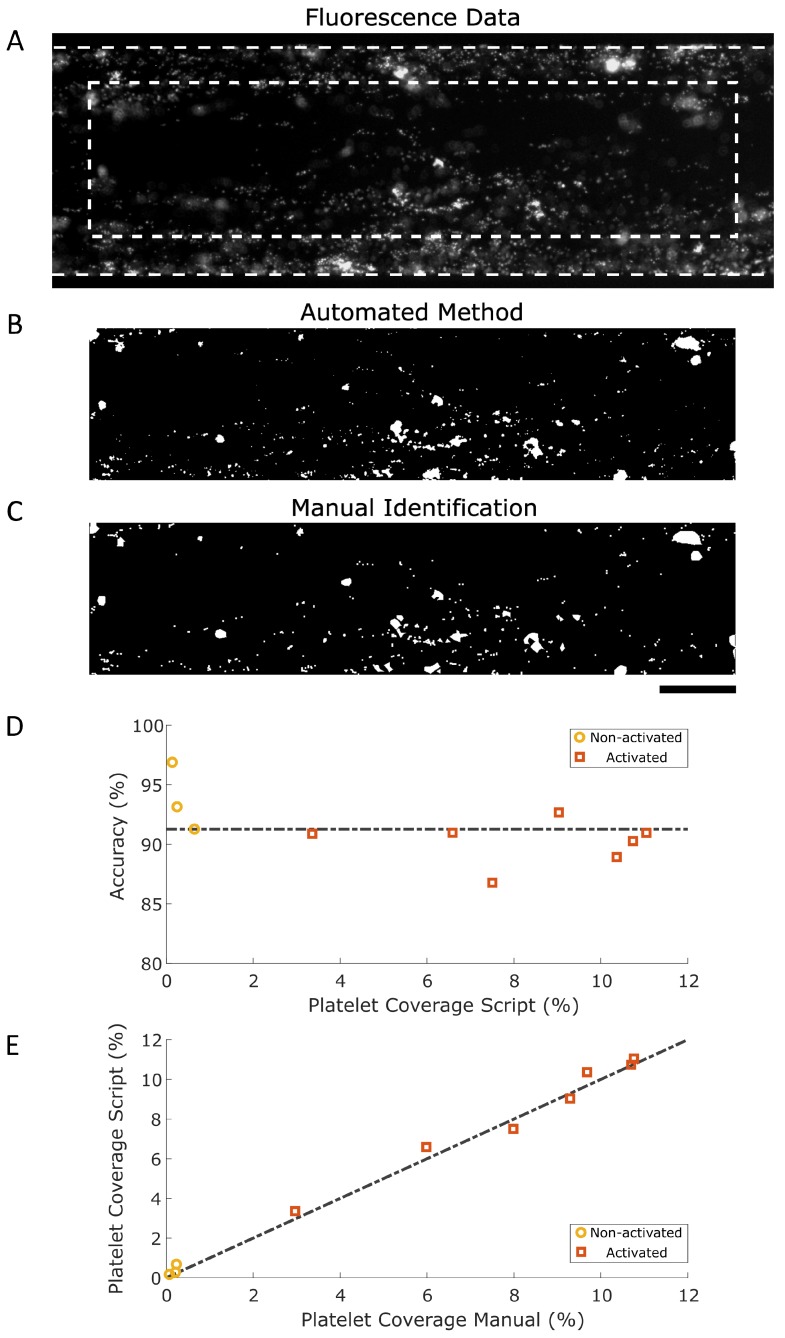
Comparison of fluorescence data of platelet aggregates processed using the automated method and manual identification. (**A**) Raw data showing platelet aggregates in greyscale. Dashed lines show the channel walls and ROI #2. ROI #1 spans the entire microfluidic channel, and ROI #2 is the area of the microfluidic channel minus the area with a 50 µm outer rim. (**B**) Binary image produced using the triangle method. (**C**) Binary image produced using the manual method. (**D**) Accuracy versus platelet coverage determined using the triangle methodology. Black dashed line represents mean accuracy. Both the activated (orange squares) and non-activated (yellow circles) endothelium are displayed. (**E**) Platelet coverage using the triangle method versus using the manual method. Black dashed function represents the line where the platelet coverage determined using the manual method correlates perfectly with the triangle method (*y* = *x*). Both the activated (orange squares) and non-activated (yellow circles) endothelium are displayed in the graph, showing a lower platelet coverage for the untreated endothelium in comparison to that for the inflamed endothelium. Scale bar, 100 µm.

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
