# Peer review of "Automated Analysis of Platelet Aggregation on Cultured Endothelium in a Microfluidic Chip Perfused with Human Whole Blood"

_micromachines, 2019, doi:10.3390/mi10110781_

Round 1

Reviewer 1 Report

The authors describe the use of a simple endothelialized microfluidic chip to quantify platelet adhesion using an automated “triangle method” versus manual quantification of platelets and platlet aggregates. The authors find that the triangle method works well and can identify more accurately the number of adhered platelets to the endothelium when the platelets are fluorescently tagged. The manuscript mostly well written in terms of grammar. However, there are a number of items in the manuscript that are not specified or are unclear. Moreover, this manuscript is essentially one brief study and fails to discuss the importance of the work described.

Comments:

Introduction, Page 2, Line 53. The average reader of this journal will likely not know what mean, Otsu, and triangle automated threshold techniques are. A very brief description of these would be useful to include.

Figure 1A does not appear to be referenced in the text.

How were the cells seeded on the top of the device? Was the device turned over? What about the side walls of the channel?

Page 2, Line 88. “…for an overnight” should be changed to “…and incubating overnight” for clarity and to be grammatically correct.

How was the arterial shear rate of 1000 s^-1 calculated or verified?

Are the devices sterilized after fabrication? This seems like a potential point where any bacteria growth could impact subsequent studies.

The methods description of the cell-staining/labeling should potentially be reworked. The methods section is not clear on which cells are being stained with what. Moreover, Figure 1 suggests that the device with endothelial cells is stained and imaged separately from when labeled platelets are introduced. This is not at all clear from the text. I would advise breaking section 2.1 in to several smaller sections such as 2.1 Device Fabrication, 2.2. Cell culture and endothelial cell imaging, 2.3. Whole blood perfusion and microscopy.

I’m not sure that this device truly constitutes an “organ-on-a-chip” given that only a single cell type is used. Endothelium-on-a-chip would be more accurate.

Figure 1 E and F. Addition of small arrows would be useful to point to aggregates.

Is TNF-alpha the endothelial cell activator? This is not clearly stated. If it is, then it should be stated in the methods that both this group and a non-treated inactivated group was employed.

There are no statistics described. The comparison between manual and automated should be considered using statistics to determine how significant the differences between these methods are.

Was the automated image produced by the triangle method of the DiOC6 stained cells compared to manual analysis? It seems like this comparison would be a natural piece of data relevant to this work.

Author Response

Reviewer 1.

“The authors describe the use of a simple endothelialized microfluidic chip to quantify platelet adhesion using an automated “triangle method” versus manual quantification of platelets and platlet aggregates. The authors find that the triangle method works well and can identify more accurately the number of adhered platelets to the endothelium when the platelets are fluorescently tagged. The manuscript mostly well written in terms of grammar. However, there are a number of items in the manuscript that are not specified or are unclear. Moreover, this manuscript is essentially one brief study and fails to discuss the importance of the work described.”

We would like to thank the reviewer for the comments. In the section below we will address the expressed concerns and suggestions in a point by point manner. Regarding the reviewer’s comment on the importance of the work, we have updated the section “Introduction” by adding the following paragraph: “Here we show for the first time that this triangle methodology is suitable for the characterization of both non-activated and activated endothelium cultured in microfluidic channels. Furthermore, the triangle methodology is compared to manual identification of single platelets and platelet aggregates and showed high degree of sensitivity, specificity and accuracy.”.

Comments 1:“Introduction, Page 2, Line 53. The average reader of this journal will likely not know what mean, Otsu, and triangle automated threshold techniques are. A very brief description of these would be useful to include.”

Reply 1:The reviewer correctly points out that we did not explain the mean, Otsu and triangle techniques in the “Introduction” section. Per suggestion of the reviewer a brief description of the mean, Otsu and Triangle technique is added: “These techniques determine a threshold by measuring the mean intensity (mean), minimizing intra-class variance in a bimodal histogram (Otsu, [25]) or by locating the base of a histogram peak (triangle, [26]).”

Comments 2:“Figure 1A does not appear to be referenced in the text. “

Reply 2: Indeed figure 1A was not mentioned in the text. We have added one line introducing Figure 1 A in the section: “Results – Microfluidic Device and Introduction of Data”: “A fluorescence microscopy image of a confluent monolayer of HUVECs is shown in Figure 1 A (nuclei in blue, F-actin in green)”

Comments 3: “How were the cells seeded on the top of the device? Was the device turned over? What about the side walls of the channel? “

Reply 3: Details about the cell seeding procedure are given in the Materials and Methods section. Endothelial cells were seeded twice. After the first incubation step, a second volume of endothelial cell suspension was introduced in the microfluidic channels and channels were incubated upside down.

The image analysis methodology omits the region closer than 50 µm from the walls, because of edge effects, thus endothelial cells do not need to cover the outer walls. However, these channel walls are merely 51 µm high (the equivalent of a few endothelial cells), and can easily proliferate onto the channel walls from the bottom or top of the channel. This was demonstrated previously (Jain et al.) and we now included a reference to that earlier paper in our M&M section - Jain, A., van der Meer, A. D., Papa, A. L., Barrile, R., Lai, A., Schlechter, B. L., ... & Frelinger, A. L. (2016). Assessment of whole blood thrombosis in a microfluidic device lined by fixed human endothelium. Biomedical microdevices, 18(4), 73.

Comments 4: “Page 2, Line 88. “…for an overnight” should be changed to “…and incubating overnight” for clarity and to be grammatically correct. “

Reply 4: We have changed the sentence in the section “Materials and Methods – Endothelial Cell Seeding and Introduction of Inflammatory Cytokines” to “… and incubating overnight”.

Comments 5: “How was the arterial shear rate of 1000 s^-1 calculated or verified? “

Reply 5: The mean wall shear rate was calculated using the following equation:

y = 6Q / wh2

where γ is shear rate (s-1), Q is volumetric flow (m3/s), w is channel width (m) and h is channel height (m). Using the given channel dimensions and a shear rate of 1000 s-1. An appropriate volumetric flow is calculated and set on the syringe pumps. The aforementioned equation and variables were added to the section “Materials and Methods – Whole Blood Perfusion”.

A computational fluid dynamic simulation was conducted to investigate the shear rate profile generated using the chosen flow rate. The outer 50 µm of the microfluidic channel show lower levels of shear rate, while the inner 200 µm stays within 95% of the maximum value. These calculations lie at the basis of our chosen volumetric flows and chosen shear rates.

To increase the clarity a figure has been added to the appendix showing the wall shear rate profile in the microfluidic channel. Additionally the section “Materials and Methods – Computational Fluid Dynamics“ has been added: “COMSOL Multiphysics 5.4 was used to conduct computational fluid dynamics modeling of the wall shear rate in the microfluidic device. A section of 2 mm of the microfluidic channel was modeled using the laminar flow module for incompressible flow. A no-slip boundary condition was imposed on all walls and a volumetric flow rate of 7.8 µl/min was applied on the inlet while atmospheric pressure was maintained on the outlet. Shear rate profiles were mapped on the 3D model and the cutline data was exported to MATLAB for visualization.”

And the section “Results – Automated Threshold using the Triangle Method” was improved to explain why 50 µm edge was omitted in analysis: “…the flow rate close to the channel was affected by edge effects resulting in lower wall shear rates. The area spanned by ROI#2 shows a flattened wall shear rate profile and displays shear rates ≥95% of the maximum (Appendix Figure A1).”

Comments 6: “Are the devices sterilized after fabrication? This seems like a potential point where any bacteria growth could impact subsequent studies. “

Reply 6: The microfluidic devices were bonded to glass microscopy slides after a brief exposure to a plasma, resulting in hydrophilic surfaces (necessary for bonding) and sterilization (due to UV exposure during the treatment). After bonding the microfluidic devices were only handled in sterile environments, minimizing the risk of bacterial infections. To clarify the section “Materials and Methods – Microfluidic Device Fabrication” the sentence describing the bonding and sterilization was changed into: “The PDMS was bonded to a glass slide (ThermoFisher) using a plasma treatment (50W, 50 kHz, 0.5 Torr for 40 seconds, CUTE MPR, Femto Science), which resulted in a sterile microfluidic channel of 51 µm × 300 µm × 14 mm (Figure 1 B)”

Comments 7: “The methods description of the cell-staining/labeling should potentially be reworked. The methods section is not clear on which cells are being stained with what. Moreover, Figure 1 suggests that the device with endothelial cells is stained and imaged separately from when labeled platelets are introduced. This is not at all clear from the text. I would advise breaking section 2.1 in to several smaller sections such as 2.1 Device Fabrication, 2.2. Cell culture and endothelial cell imaging, 2.3. Whole blood perfusion and microscopy. “

Reply 7:  We thank the reviewer for the suggestions regarding the section “Materials and Methods”. The first subsection is now divided into three separate sections: “Microfluidic Device Fabrication”, “Endothelial Cell Seeding and Introduction of Inflammatory Cytokines” and “Whole Blood Perfusion”. To improve clarity, the section describing endothelial cell staining is now moved to the section “Endothelial Cell Seeding and Introduction of Inflammatory Cytokines” and is separate from the section describing labeled platelets.

Comments 8: “I’m not sure that this device truly constitutes an “organ-on-a-chip” given that only a single cell type is used. Endothelium-on-a-chip would be more accurate. “

Reply 8: In the section “Materials and Methods” we have changed the first subtitle into “Microfluidic Device Fabrication”. All other mentions of organ-on-a-chip devices in the manuscript refer to work conducted by others or new possible applications for the methodology.

Comments 9: “Figure 1 E and F. Addition of small arrows would be useful to point to aggregates. “

Reply 9: We agree with the reviewer that the addition of small arrows in the insets would improve clarity. Thus Figure 1 has been updated to include arrows indicating a platelet aggregate and individual platelets.

Comments 10: “Is TNF-alpha the endothelial cell activator? This is not clearly stated. If it is, then it should be stated in the methods that both this group and a non-treated inactivated group was employed. “

Reply 10: Indeed TNF-α is used to induce inflammation and both inflamed and untreated endothelium are mentioned in the results. The reviewer correctly points out that this is not mentioned in the section “Material and Methods”. Therefore, the section “Materials and Methods - Endothelial Cell Seeding and Introduction of Inflammatory Cytokines” is updated: “Cells were allowed to reach confluency before introducing ECGM with (activated condition) or without (non-activated condition) 10-50 ng/ml Tumor Necrosis Factor-α (TNF-α, Sigma Aldrich) and incubating overnight”

Comments 11: “There are no statistics described. The comparison between manual and automated should be considered using statistics to determine how significant the differences between these methods are. “

Reply 11:Indeed no statistics were present in the original manuscript. We have investigated the overlap between the mask generated by the triangle methodology and the mask resulting from manual identification of platelet aggregates. This resulted in numbers on true positives, false positives, true negatives and false negatives and sensitivity, specificity and accuracy to be calculated.

We have added the “Materials and Methods – Manual Identification of Platelet Aggregates and Statistics”  section: “ROI #2 was manually processed by identifying individual platelets and platelet aggregates in MATLAB. The black and white images resulting from the manual identification of platelets and platelet aggregates were compared to the black and white images from the triangle methodology. A direct comparison resulted in true positives, false positives, true negatives and false negatives used to calculate sensitivity, specificity and accuracy. After calculating platelet coverages, using the same method as mentioned in section 2.4, the platelet coverages found with the triangle methodology were plotted versus the manual identification method. Furthermore, the R-squared value was determined, by calculating squared residuals compared to y = x using MATLAB.”

Furthermore, we have updated the section “Results - Comparison of Binary Output using the Triangle Method with the Manual Method” to add: “To determine how the triangle methodology compares to manual identification, masks from both techniques were superimposed and the statistical metrics sensitivity = 84.84%, specificity = 98.69% and accuracy = 91.27% were determined. The triangle methodology recognized true positive values (sensitivity) and was even better at identifying true negatives (specificity). Furthermore, the triangle methodology was capable to detect both true positives and true negatives resulting in a high accuracy.” and have updated Figure 4.

Comments 12: “Was the automated image produced by the triangle method of the DiOC6 stained cells compared to manual analysis? It seems like this comparison would be a natural piece of data relevant to this work.”

Reply 12: The DiOC6 data was included to show that the methodology also works for labels with a higher background signal. We have conducted the manual identification of platelet aggregates and added the platelet coverages measured by the script and by manual identification to section: “Appendix”: “Binary images produced using the triangle method, which resulted in platelet coverages of 6.526% (C) and 4.651% (D). Manual identification of platelets and platelet aggregates resulted in platelet coverages of 6.654% (C) and 4.597% (D).”

Reviewer 2 Report

This work by Albers et al presents the use of the triangle method for segmenting images of aggregated and non-aggregated platelets in a microfluidic channel. 

The authors coat the channel with activated or non-activated endothelial cells, and fluorescently stain platelets after flowing blood through this channel. To detect platelets, the image is first rotated to align the channel in a horizontal position. Afterwards, an ROI of the channel is determined to calculate the threshold according to the triangle method. This threshold is then applied to a smaller ROI, 50 um inward of the first RO1. The authors compare the results to manual annotation by comparing the platelet coverage (percentage of area covered by platelets). 

The authors fail to provide enough background regarding the microfluidic device or the triangle method. This method is well known and was developed over 40 years ago, and is available in multiple image processing software (ImageJ, MATLAB). In addition to a lack of novelty, the following concerns were identified:

The comparison of manual annotation vs image processing is not appropriate. The metrics used only determine whether both algorithms provide a similar total area identified, but these areas could be completely non-overlappping. The authors should use the well-known metrics of accuracy, true positive, false negative, etc.  The authors state that single platelets cannot be annotated by hand. According to figure 4B they are visible and thus can be annotated by hand.  Appendix A: these histograms have the same problem as Figure 4. These metrics are not good indicators for code performance. It appears as if the rotation and ROIs are done manually, but could easily be done via a script. The use of a 50 um smaller ROI is unclear, why are 50 um enough? This is an arbitrary number, justification of this smaller ROI is necessary It would be relevant to compare performance with other methods, such as Otsu's, k-means Explanation as to why the triangle method works (and others don't) in this particular scenario is necessary "The angle alpha indicates skewness" I would suggest changing the word 'skewness' The authors write the entire paper as if the only important part is the image processing. I would suggest emphasizing the microfluidic device and potentially running more experiments to show this device can be useful in conjunction with the image processing.

Author Response

Reviewer 2.

“This work by Albers et al presents the use of the triangle method for segmenting images of aggregated and non-aggregated platelets in a microfluidic channel. 

The authors coat the channel with activated or non-activated endothelial cells, and fluorescently stain platelets after flowing blood through this channel. To detect platelets, the image is first rotated to align the channel in a horizontal position. Afterwards, an ROI of the channel is determined to calculate the threshold according to the triangle method. This threshold is then applied to a smaller ROI, 50 um inward of the first RO1. The authors compare the results to manual annotation by comparing the platelet coverage (percentage of area covered by platelets). 

The authors fail to provide enough background regarding the microfluidic device or the triangle method. This method is well known and was developed over 40 years ago, and is available in multiple image processing software (ImageJ, MATLAB). In addition to a lack of novelty, the following concerns were identified:”

We would like to thank the reviewer for the comments. In the section below we will address the expressed concerns and suggestions in a point by point manner.

Comments 1:“The comparison of manual annotation vs image processing is not appropriate. The metrics used only determine whether both algorithms provide a similar total area identified, but these areas could be completely non-overlappping. The authors should use the well-known metrics of accuracy, true positive, false negative, etc.  “

Reply 1:Indeed no statistics were present in the original manuscript. We have investigated the overlap between the mask generated by the triangle methodology and the mask resulting from manual identification of platelet aggregates (including single platelets as suggested in next comment). This resulted in numbers on true positives, false positives, true negatives and false negatives and sensitivity, specificity and accuracy to be calculated.

We have added the “Materials and Methods – Manual Identification of Platelet Aggregates and Statistics”  section: “ROI #2 was manually processed by identifying individual platelets and platelet aggregates in MATLAB. The black and white images resulting from the manual identification of platelets and platelet aggregates were compared to the black and white images from the triangle methodology. A direct comparison resulted in true positives, false positives, true negatives and false negatives used to calculate sensitivity, specificity and accuracy. After calculating platelet coverages, using the same method as mentioned in section 2.4, the platelet coverages found with the triangle methodology were plotted versus the manual identification method. Furthermore, the R-squared value was determined, by calculating squared residuals compared to y = x using MATLAB.”

Furthermore, we have updated the section “Results - Comparison of Binary Output using the Triangle Method with the Manual Method” to add: “To determine how the triangle methodology compares to manual identification, masks from both techniques were superimposed and the statistical metrics sensitivity = 84.84%, specificity = 98.69% and accuracy = 91.27% were determined. The triangle methodology recognized true positive values (sensitivity) and was even better at identifying true negatives (specificity). Furthermore, the triangle methodology was capable to detect both true positives and true negatives resulting in a high accuracy.” and have updated Figure 4.

Comments 2:“The authors state that single platelets cannot be annotated by hand. According to figure 4B they are visible and thus can be annotated by hand.  Appendix A: these histograms have the same problem as Figure 4. These metrics are not good indicators for code performance. “

Reply 2:The reviewer is right that small aggregates and individual platelets are visible in figure 4B and can thus be manually identified. We have added a higher level of detail to all data presented in the manuscript, also including single platelet features and the data shown in figure 4D has been updated. We thank the reviewer for the suggestion, since the difference between the manual method and the triangle methodology was mainly comprised of small aggregates and has dwindled.

Comments 3:“It appears as if the rotation and ROIs are done manually, but could easily be done via a script. “

Reply 3:The top edge of the microfluidic channel is indeed manually indicated in a MATLAB UI, but the angle correction and positioning of the ROIs are all done automatically. We have added the option to automatically detect channel edge, correct for misalignment and define ROIs to the MATLAB script and have updated the section: “Results – Automated Threshold using the Triangle Method” with the following sentence: “Alternatively, the channel edge was found automatically by vertically scanning the image to find the first local maximum  intensity followed by an angle sweep to determine the misalignment angle.”

Comments 4:“The use of a 50 um smaller ROI is unclear, why are 50 um enough? This is an arbitrary number, justification of this smaller ROI is necessary “

Reply 4:A computational fluid dynamic simulation was conducted to investigate the shear rate profile generated using the chosen flow rate. The outer 50 µm of the microfluidic channel show lower levels of shear rate, while the inner 200 µm stays within 95% of the maximum value.

To increase the clarity and to make sure this is clear for any future reader a figure has been added to the appendix showing the wall shear rate profile in the microfluidic channel. Additionally the section “Materials and Methods – Computational Fluid Dynamics“ has been added: “COMSOL Multiphysics 5.4 was used to conduct computational fluid dynamics modeling of the wall shear rate in the microfluidic device. A section of 2 mm of the microfluidic channel was modeled using the laminar flow module for incompressible flow. A no-slip boundary condition was imposed on all walls and a volumetric flow rate of 7.8 µl/min was applied on the inlet while atmospheric pressure was maintained on the outlet. Shear rate profiles were mapped on the 3D model and the cutline data was exported to MATLAB for visualization.”

And the section “Results – Automated Threshold using the Triangle Method” was improved to explain why 50 µm edge was omitted in analysis: “…the flow rate close to the channel was affected by edge effects resulting in lower wall shear rates. The area spanned by ROI#2 shows a flattened wall shear rate profile and displays shear rates ≥95% of the maximum (Appendix Figure A1).”

Comments 5: “It would be relevant to compare performance with other methods, such as Otsu's, k-means Explanation as to why the triangle method works (and others don't) in this particular scenario is necessary “

Reply 5:The section “Results – Microfluidic Device and Introduction of Data” gives some background into why the triangle technique is favourable, and explains how a selection of techniques was tested in ImageJ to find a suitable candidate. Both Otsu and Triangle seemed suitable candidates, but to further explain why we favoured the triangle technique we have added the following sentence to the section “Results – Microfluidic Device and Introduction of Data”: “However, minimal platelet adhesion is expected on non-activated endothelium, resulting in a unimodal histogram making the Otsu technique less suitable.”

Comments 6:“"The angle alpha indicates skewness" I would suggest changing the word 'skewness’”

Reply 6:We have updated the term “skewness” to channel misalignment.

Comments 7:“The authors write the entire paper as if the only important part is the image processing. I would suggest emphasizing the microfluidic device and potentially running more experiments to show this device can be useful in conjunction with the image processing. “

Reply 7:The communication paper was indeed written with an emphasis on the image analysis, which was tested with data collected from microfluidic devices. The communication format allows authors to submit promising preliminary results or single issues. The main results that we would like to publish is that we are the first to successfully apply the triangle methodology to characterize platelet aggregation in endothelialized microfluidic devices. This is important, particularly because these devices present widely varying expected platelet coverages, as opposed to miniaturized flow chambers with uniform coating of proteins. Furthermore, we have investigated the performance of the methodology and compared results with manual identification of platelet aggregates using well-known metrics per your suggestions.

To better indicate the main message, additions were made to the section “Introduction”: “Here we show for the first time that this triangle methodology is suitable for the characterization of both non-activated and activated endothelium cultured in microfluidic channels. Furthermore, the triangle methodology is compared to manual identification of single platelets and platelet aggregates and showed high degree of sensitivity, specificity and accuracy.”

Furthermore, changes were made to the section “Conclusions”: “In this communication an automated analysis of platelet coverage and platelet size distribution was introduced for applications in organs-on-a-chips and vasculature-on-a-chip devices. The methodology corrects for channel misalignment, automatically defines regions of interest and sets a threshold using the triangle method. The described method was compared to a manual identification method, where a user manually indicated adhered platelets and aggregates. Platelet coverages determined by both techniques showed a high degree of correlation. Furthermore, a high sensitivity, specificity and accuracy were measured. The image analysis method presented here is capable of determining platelet coverages and platelet size distributions in microfluidic devices perfused with human whole blood lined with either activated or untreated endothelial cells proving the robustness of the methodology.”

Reviewer 3 Report

The authors in the current article present a image analysis based method for automated analysis of platelet coverage and size. The steps adapted to correct skewness, identify region of interest etc. are well described. The manuscript is written in a well organized fashion with adequate details where necessary. Figures and images are well formatted with descriptive captions. 

Recommend accepting the manuscript for publication in the journal.

Author Response

We would like to thank the reviewer for the comments, a positive recommendation and for taking the time to review the manuscript.

Round 2

Reviewer 1 Report

The authors have sufficiently addressed the reviewer comments and have revised the manuscript accordingly.

Reviewer 2 Report

The authors have addressed my concerns to some degree. The additional performance metrics (sensitivity, specificity, accuracy) certainly show that the image processing algorithm is successful at detecting platelets and aggregates. Most of my concerns have been addressed, however, I still find Figure 4D problematic. This correlation has no meaning, unless I am not understanding what this plot signifies. From what I understand, this graph says wether the amount of pixels detected by the triangle method is the same as the amount of pixels detected manually. This does not take into account pixel location, the fact that both methods identify a similar amount of pixels is an illogical performance metric. It has a very high correlation, but using this as a metric for performance of the code seems misleading. If I am understanding this figure incorrectly, then I think this needs further explanation in the maint ext.